
# Surveying the Surveyors to Address Risk Perception and Adaptive Behaviour Cross-study Comparability

Samuel Rufat[1,2], Mariana Madruga de Brito[3], Alexander Fekete[4], Emeline Comby[5], Peter J. Robinson[6], Iuliana Armaș[7], Wouter J.W. Botzen[8], Christian Kuhlicke[3,9]

[1]CY Cergy Paris University, 95011, Cergy-Pontoise, France. ORCID 0000-0001-6356-1233
[2]Institut Universitaire de France, 75005, Paris, France.
[3]Department of Urban and Environmental Sociology, Helmholtz Centre for Environmental Research, Leipzig, 04318, Germany, M.M.d.B. ORCID 0000-0003-4191-1647
[4]Institute of Rescue Engineering and Civil Protection, TH Köln – University of Applied Sciences, Betzdorferstr. 2, 50679
Cologne, Germany, ORCID 0000-0002-8029-6774
[5]University Lumière Lyon 2, UMR 5600 EVS CNRS, Lyon, 69007, France, ORCID 0000-0003-4057-3623
[6]Department of Environmental Economics, Institute for Environmental Studies (IVM), VU University Amsterdam, De Boelelaan 1111, 1081 HV Amsterdam, The Netherlands. ORCID 0000-0003-2833-8030
[7]University of Bucharest, 010041, Bucharest, Romania. ORCID 0000-0002-8020-6767
[8]Department of Environmental Economics, Institute for Environmental Studies (IVM), VU University Amsterdam, De Boelelaan 1111, 1081 HV Amsterdam, The Netherlands. ORCID 0000-0002-8563-4963
[9]Institute for Environmental Sciences and Geography, University of Potsdam, 14468 Potsdam-Golm, Germany. ORCID 0000-0002-1193-228X

*Correspondence to*: Samuel Rufat (samuel.rufat@u-cergy.fr), Mariana Madruga de Brito (mariana.brito@ufz.de)

**Abstract.** One of the key challenges for risk, vulnerability, and resilience research is how to address the role of risk perceptions and how perceptions influence behaviour. It remains unclear why people fail to act adaptively to reduce future losses, even when there is ever richer information available on natural and human-made hazards (flood, drought, etc.). The current fragmentation of the field makes it an uphill battle to cross-validate the results of existing independent case studies. This, in turn, hinders comparability and transferability across scales and contexts and hampers recommendations for policy

and risk management. To improve the ability of researchers in the field to work together and build cumulative knowledge, we question whether we could agree on (1) a common list of minimal requirements to compare studies, (2) shared criteria to address context-specific aspects of countries and regions, and (3) a selection of questions allowing for comparability and long-term monitoring. To map current research practices and move in this direction, we conducted an international survey – the Risk Perception and Behaviour Survey of Surveyors (Risk-SoS). We find that most studies are exploratory in nature and

often overlook theoretical efforts that would enable the comparison of results and an accumulation of evidence. While the diversity of approaches is an asset, the robustness of methods is an investment. Surveyors report a tendency to reproduce past research design choices but express frustration with this trend, hinting at a turning point. To bridge the persisting gaps, we offer several recommendations for future studies, particularly grounding research design in theory, improving the formalisation of methods, and formally comparing theories and constructs, methods and explanations while collecting the

most-in-use themes and variables and controlling for the most-in-use explanations.



# 1 Introduction

One of the key challenges for risk, vulnerability, and resilience studies is understanding risk perceptions and how these perceptions influence behaviour. A central question is why people fail to act adaptively to reduce future losses, even when there is increasingly richer risk information provided by various communication channels (e.g., websites, social media, mobile applications, television and print news). Whilst United Nations (UN) programs aim to foster public engagement and community participation in disaster preparedness, recovery and adaptation (UNDRR, 2015; 2019), we have a fragmented understanding of risk perception and risk reduction behaviour drivers (Lechowska, 2018). The current focus of risk management on structural measures, monetary impacts and cost-benefit analyses frequently relies on flawed underlying assumptions, as they leave aside social inequalities, actual behaviour, underlying motivations and capacities that can lead to significant differences in resilience across society (Rufat et al., 2020; Kuhlicke et al., 2020). Such a narrow focus runs the risk of hollowing out resilience by overlooking citizens' perceptions, knowledge, capacities, motivations, and behaviours. This hinders the achievement of more inclusive climate change adaptation (CCA) and disaster risk reduction (DRR) called for by the UN Sendai Framework (2015–2030) and the UN Sustainable Development Goals (SDGs 2030).

The current fragmentation of academic research fields on risk perception, behaviour and adaptation, and the historically disparate development of DRR and CCA communities, hinder comparability and transferability across scales and contexts in research fields defined by high degrees of uncertainty. The interdisciplinary theories and methods used are shaped by different sets of assumptions and often lead to contradictory findings (Bradford et al. 2012). Competing theories and divergent methods fragment our understanding of risk perception (Bamberg et al. 2017), with disagreement on drivers (Lechowska, 2018) and their interactions (Rufat et al., 2015) and influence on individual behaviour (Bubeck et al., 2012). Whilst predicting the actual behaviour of people before, during and after a crisis remains a major challenge (Kreibich et al., 2017), it is often assumed that risk communication and awareness campaigns can foster desired judgement, motivation, and behaviour (Rufat et al., 2020). Most theories assume that high-risk perception will lead to personal preparedness and then to risk mitigation behaviour, but it has been verified repeatedly that high risk perceptions do not lead to preparedness or adaptive action (Wachinger et al., 2013). The current 'behavioural turn' in DRR and CCA (Kuhlicke et al., 2020) overlooks this gap, with recent strategies advocating that less protected households are individually responsible for looking after themselves. The reasons for this shift are that stretched public budgets are deemed unable to carry the costs for upgrading structural measures (Slavikova 2018) and policy is increasingly relying on individual resilience (Begg et al., 2017).

The main sources of uncertainty in the design of risk perception, behaviour and adaptation studies include the many drivers of risk reduction behaviour, demographic, social and cultural factors (Wilkinson, 2001; de Brito et al., 2018), under- or overestimation of risk (Mol et al., 2020), place attachment (de Dominicis et al., 2015), previous hazard experience (Botzen et al., 2015) or the use of short-term horizons by households and decision-makers in planning and risk management (Hartmann



et al., 2017). However, it remains challenging to disentangle which factors drive risk perception in a specific area or among specific groups (Rufat, 2015).

Diverging risk perception and behaviour theories are used in studies to test limited sets of hypotheses, drivers or control variables, resulting in findings that are not easily rendered compatible (Lechowska, 2018). Although numerous theoretical frameworks have been developed of ways in which risk perceptions are formed and relate to preparedness and/or adaptive behaviour (e.g., Kellens et al., 2013; Robinson and Botzen, 2019; van Valkengoed and Steg, 2019), no definitive explanation has yet been found (Siegrist et al. 2020), and opposite conclusions can be reached from different case studies (Wachinger et al., 2013). Existing theories focus on different dimensions (sociological, economical, psychological), internal or personal factors (gender, age, education, income, values, trust), external or contextual factors (e.g., vulnerability, institutions, power, oppression or cultural backgrounds), risk or environmental factors (e.g., perceived likelihood or experienced frequency), and/or informational factors (e.g., media coverage, experts or risk management). This situation is not satisfying in the long run, as it hinders the production of a common baseline for risk perception and adaptation studies and prevents the comparison of empirical insights derived from different studies and thus the accumulation of evidence. The current fragmentation of the field makes it an uphill battle to cross-validate the results of the current collection of independent case studies. This, in turn, hinders comparability and transferability across scales and contexts and hampers recommendations for policy and risk management. Improving comparability would significantly increase the ability of researchers from different communities to work together and build cumulative knowledge.

By sending a *Survey of Surveyors* (SoS) to the research community, we wanted to initiate a discussion on research standards in this field. While we obviously cannot all run the same questionnaire or focus groups – because we have different research questions, case studies, geographical settings and social contexts – our ability to work together and build cumulative knowledge can be significantly improved by having: (1) a common list of minimal requirements to compare studies and surveys, (2) a set of shared criteria to address context-specific aspects of countries and regions and (3) a selection of survey questions or themes allowing for comparability and long-term monitoring. We conducted an international survey aiming to map current research practices. The *Risk Perception and Behaviour Survey of Surveyors* (Risk-SoS) intended to foster convergence and comparison in risk perception, behaviour and adaptation studies. More specifically, we wanted to investigate which theories, variables and elements are frequently targeted by surveyors.

## 2 Methods, questionnaire and dissemination

With this survey, we aimed to identify core elements for enabling results comparability while allowing individual surveys to pursue their other specific questions. The original discussion started at the first European Conference on Risk Perception, where struggles to define these core elements within a limited group of experts around the table occurred (Rufat and Fekete, 2019). Therefore, the idea of this Risk-SoS was born.





The survey consisted of 30 questions, mainly multiple-choice questions (see supplementary material). Established brainstorming techniques were used during webinar group discussions to select the questions to be included. The first three questions dealt with the respondents' methodological practices in terms of data collection. Questions 4-7 focused on the disciplines and social theories used by the surveyors. Questions 8-14 addressed the variables analysed, focusing on explanatory variables such as age, gender and education. Questions 15-19 related to the pre-pandemic and post-pandemic survey designs and sample sizes. Questions 20-22 discussed the comparison effort and expectations in terms of the variables compared. The final questions described the surveyor's experience in terms of diversity of case studies and risks studied; they also captured demographic variables (country of residence, gender, employment and education). Before disseminating the survey, we tested it within our group to eliminate ambiguities.

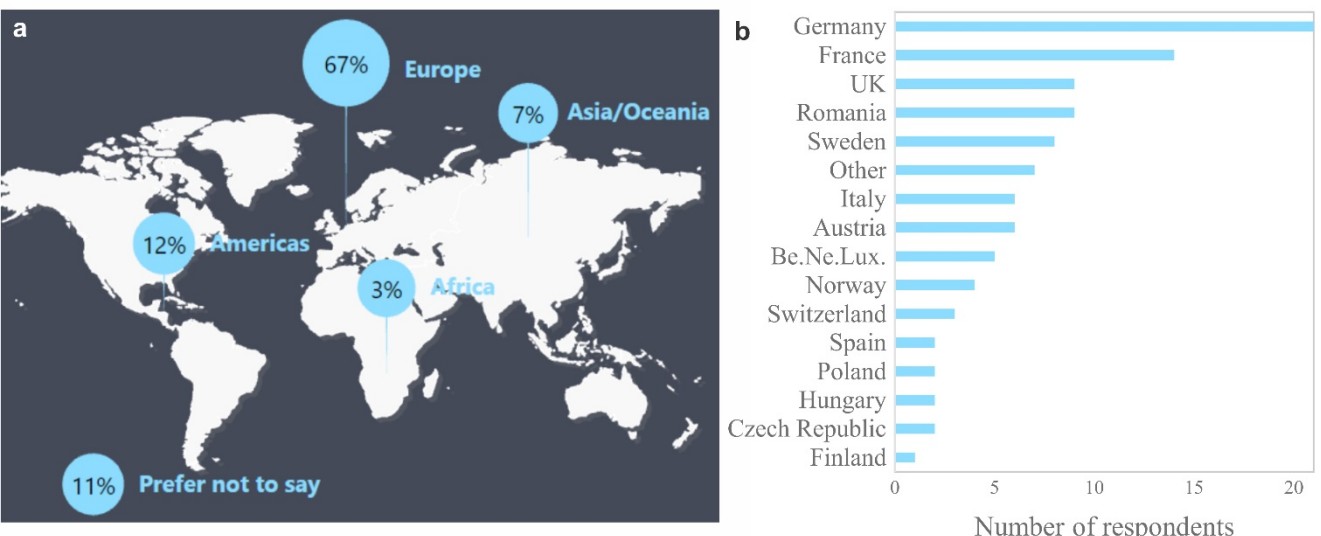

Figure 1: (a) Continent of employment of the 150 respondents, (b) Number of respondents according to the participant's country of employment in Europe. Be.Ne.Lux corresponds to Belgium, Netherlands and Luxembourg.

The Risk-SoS survey was disseminated in a snowball fashion to reach the community by sending personal emails between December 2020 and April 2021. We first wrote personally to scientists who had published empirical studies in English academic journals relevant to the field during the last 20 years. We wrote to colleagues based in Europe, requesting them to forward the survey invitation to other persons who might be interested, and subsequently sent several reminders. Although our original focus was on Europe, the snowball effect allowed us to reach other continents. One hundred and fifty experts from more than 25 countries, one quarter outside of Europe (Fig. 1), answered the survey ($N = 150$). Their backgrounds included experience in individual or community perceptions of risk, climate impacts or hazards adaptation behaviour, using surveys, interviews, experiments or focus groups.

Our sample was balanced in terms of gender (Table 1). Most surveyors ($n = 107$, 71%) had a PhD, some of the others were PhD students. The snowball method allowed us to reach a population with substantial experience in research. This observation was confirmed by the number of studies already conducted. Just under half of the respondents ($n = 68$, 45%) had conducted more





than three risk studies, 53 (35%) had conducted one or two studies, and 13 (9%) were currently working on their first study. Most respondents (*n* = 114, 76%) were currently working in academia. Thus, our respondents had experience with risk studies. A minority of the respondents (*n* = 42, 28%) did not wish to be associated with only one humanities and social sciences discipline. A quarter of the respondents considered themselves geographers (*n* = 38, 25%), 18 (13%) as environmental scientists, 14 (9%) as sociologists, 12 (8%) as psychologists and 10 (7%) as economists. The results were collected and treated anonymously. They were shared and discussed with the community during monthly Risk-SoS webinars (Rufat et al., 2021).

| Gender | n | % | | n | % |
|---|---|---|---|---|---|
| Female | 67 | 45 | Prefer not to say | 19 | 13 |
| Male | 63 | 42 | Other | 1 | 2 |
| **Main field of PhD or studies** | n | % | | n | % |
| Geography | 38 | 25 | Economy | 10 | 7 |
| Other | 24 | 16 | Political science | 7 | 5 |
| Environment | 19 | 13 | Anthropology | 3 | 2 |
| Prefer not to say | 18 | 12 | Management | 3 | 2 |
| Sociology | 14 | 9 | Communication | 2 | 2 |
| Psychology | 12 | 8 | | | |
| **Years since PhD** | n | % | | n | % |
| Student or no PhD | 30 | 20 | 13 to 20 years | 21 | 14 |
| 1 to 3 years | 19 | 13 | Over 20 years | 18 | 12 |
| 4 to 7 years | 25 | 17 | Prefer not to say | 15 | 10 |
| 8 to 12 years | 22 | 15 | | | |
| **Work affiliation** | n | % | | n | % |
| Academia | 114 | 76 | Other | 7 | 5 |
| Prefer not to say | 12 | 8 | Think tank, consulting | 6 | 4 |
| Public service or agency | 8 | 5 | International body | 3 | 2 |
| **Number of finalised risk perception case studies** | n | % | | n | % |
| Work in progress | 13 | 9 | 11 to 20 | 6 | 4 |
| 1 or 2 | 52 | 35 | over 20 studies | 4 | 3 |
| 3 to 5 | 36 | 24 | Prefer not to say | 18 | 12 |
| 6 to 10 | 21 | 14 | | | |
| **Investigated hazards or risks** (multiple answers) | n | % | | n | % |
| Floods | 91 | 61 | Hazard agnostic, no specific hazard | 12 | 8 |
| Climate, climate change impacts | 77 | 51 | Industrial accidents | 12 | 8 |
| Earthquakes, volcanoes, landslides | 39 | 26 | Terror, military, attacks | 9 | 6 |
| Drought, extreme temperatures | 37 | 25 | Compounded, cascading events | 9 | 6 |
| Storms, cyclones, weather events | 37 | 25 | Nuclear accidents | 8 | 5 |
| Epidemics, pandemics | 34 | 23 | Traffic and transport accidents | 6 | 4 |
| Multiple hazards | 33 | 22 | Domestic accidents | 5 | 3 |
| Pollution, environmental disasters | 23 | 15 | Slow unfolding events | 4 | 3 |
| Other hazards | 19 | 13 | Protests, riots, unrest | 2 | 1 |
| Fires, wildfires | 15 | 10 | Stock market crashes, debts, recessions | 2 | 1 |
| Submersion, sea level rise | 14 | 9 | Don't know | 1 | 1 |

**Table 1: Participants' characteristics and hazards they investigate**

The study was meant to be an exploration to map current practices. Therefore, we did not *a priori* define a set of hypotheses or specify an overarching framework. A combination of descriptive statistics was used to present the results. Moreover, bivariate (Pearson correlation, chi-square test) and multivariate (logistic regression) statistics were used to assess significant relationships among the answers, as well as between replies and the respondents' backgrounds.



# 3 Theories, disciplines and frameworks used by the surveyors

Several theoretical strands from social sciences, psychology and environmental sciences have been introduced to support risk perception and behaviour studies. The use of theoretical constructs is encouraged, as they can lead to deeper and more thorough insights into the social world. Furthermore, they allow for comparison and the consequent accumulation of evidence (Kuhlicke et al., 2020; Rufat et al., 2020).

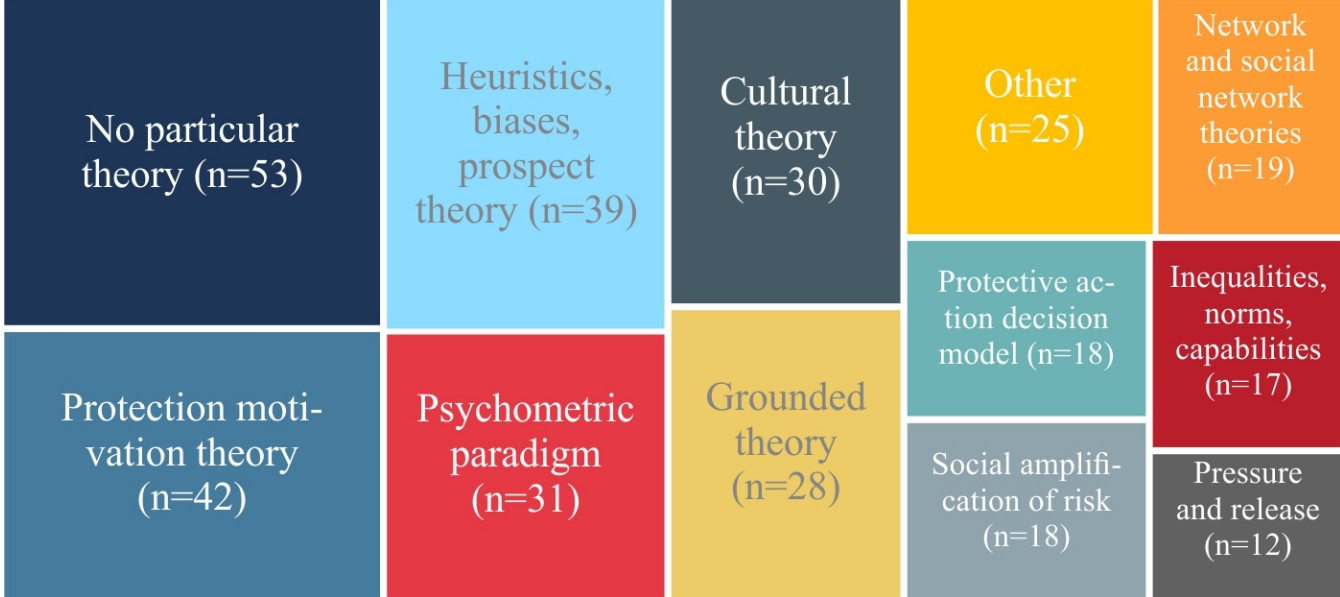

**Figure 2: Replies to the question,** *'Did you ever use a theoretical model or framework to guide the design of any of your studies on risk perception or behaviour?'* **Only theories that were mentioned more than 10 times are shown.**

Despite this importance, survey results showed that a large share of the participants (*n* = 53, 35%) had not relied on any particular theoretical model or framework to guide the design of their studies (Fig. 2). At first sight, this result might be surprising. However, it is worth mentioning that the reasons for considering theories were not captured in our survey. It may be that an underlying theory informed the research, even if the researchers did not state it clearly. Also, as many respondents mentioned that they designed their studies based on the literature or a previous study, it is possible that previous studies (including theories) inspired their choices. Additionally, it could be the case that there were good or even theoretical reasons to not apply a specific theory or to conduct a study inductively without the influence of pre-existing theories. In this regard, a participant mentioned that *'I do not tend to use a single explicit theory in a deductive way but am informed by PMT and COM-B[1]'*. The responses may have also been influenced by how the question and answers were framed. Indeed, three participants that had declared the use of *'no particular theory'* mentioned in a subsequent question that they had formally compared more than two theories in the same study.

---

1  PMT stands for Protection Motivation Theory; COM-B proposes that Behaviour consists of the components Capability, Opportunity, and Motivation.





Of those who had considered a theoretical framing, most used 'protection motivation theory' ($n$ = 42, 28%), followed by 'heuristics, biases, prospect theory', 'psychometric paradigm' and 'cultural theory'. This was expected, as these frameworks are well established in this field of research. Twenty-five (17%) participants had used other frameworks not included in the survey, such as the model of pro-environmental behaviour, the mental model approach, Cutter's framing of social

vulnerability, construal level theory, game theory, sense of place, the transtheoretical model, hyperbolic discounting and social capital, among others. The high number of 'other' responses warrants further investigation.

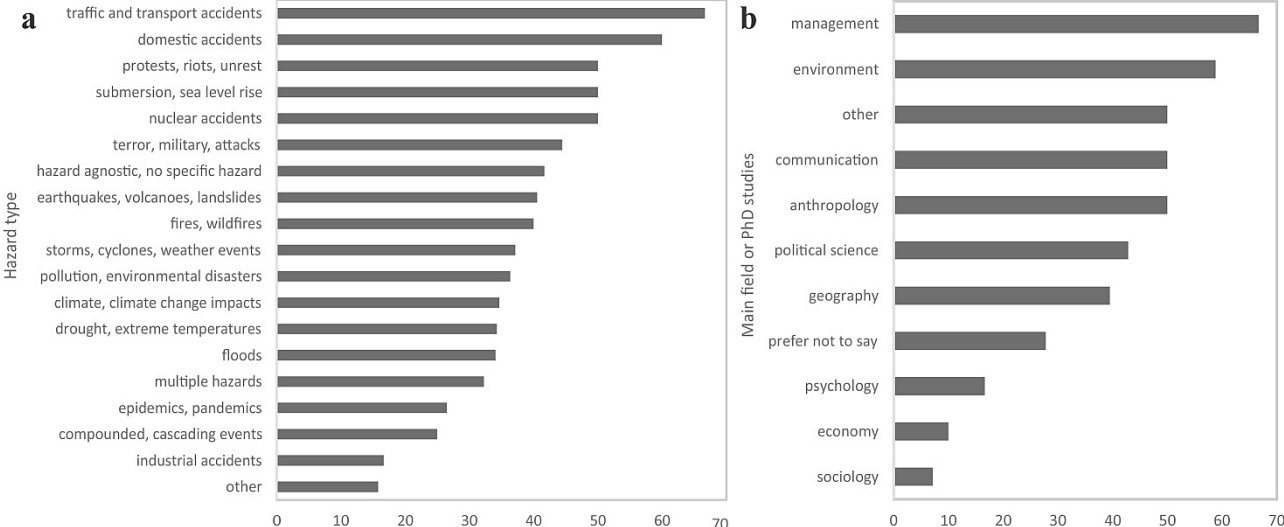

**Figure 3: Percent of respondents that did not rely on any particular theory, according to (a) the hazard they investigated and (b) their field of knowledge.**

Differences existed according to the hazard investigated (Fig. 3a). Indeed, more than 50% of the participants studying 'traffic and transport accidents', 'domestic accidents', 'protests, riots, unrest', 'sea level rise' and 'nuclear accidents' did not rely on a particular theory. Conversely, participants from the hazard fields 'industrial accidents', 'compounded, cascading events', 'epidemics, pandemics' and 'multiple hazards' tended to conduct theoretically grounded research.

Differences were also observed according to discipline. Only 7% ($n$ = 1) of the 'sociology' participants did not rely on a

particular theory. Conversely, 59% ($n$ = 10) of the participants with a background in environmental sciences did not rely on a particular theory. Given the large share of geographers ($n$ = 38, 25%) and environmental disciplines ($n$ = 18, 13%) (Table 1), it is surprising that theories from these fields (e.g., Pressure And Release, Hazards of Place) did not receive a high number of responses. However, in our results, neither discipline is particularly strong in theory application (Fig. 3b). The fact that many researchers work in interdisciplinary groups might also help to explain why standard and psychological theories had been

used more often. By grouping respondents according to their training, geographers and environmental scientists in one group, sociologists and psychologists in another, and a third group with all others, we find a significant difference in their approach to risk perception and behaviour. Those in psychology or sociology had an 85% higher chance than those in




geography or environmental disciplines to prefer a specific theory (logistic regression of -1.86, i.e. a chance of 0.155). In other words, those trained in sociology and psychology were more likely to have the methodological background to formulate

working hypotheses following specific theories and use measurement scales to capture human perceptions and behaviours.

# 4 Questions asked and themes explored by the surveyors

A key interest of this study was to identify what is being studied in risk perception surveys, which key elements are most often explicitly targeted by the surveyors and what may be deemed as out of focus. To explore this, two questions were designed to disentangle the range of choices and uncover possible convergences around key approaches and foci. While the

respondents converged around some key elements, there was less agreement on the usefulness of such convergence. Results to the question,*'What did you try to capture with your risk perception questions?',* showed that surveyors captured a multitude of elements (23 items) in their surveys (Fig. 4). This indicates that the respondents were aware that risk perception and behaviour are complex phenomena that cannot be easily reduced to a few elements.

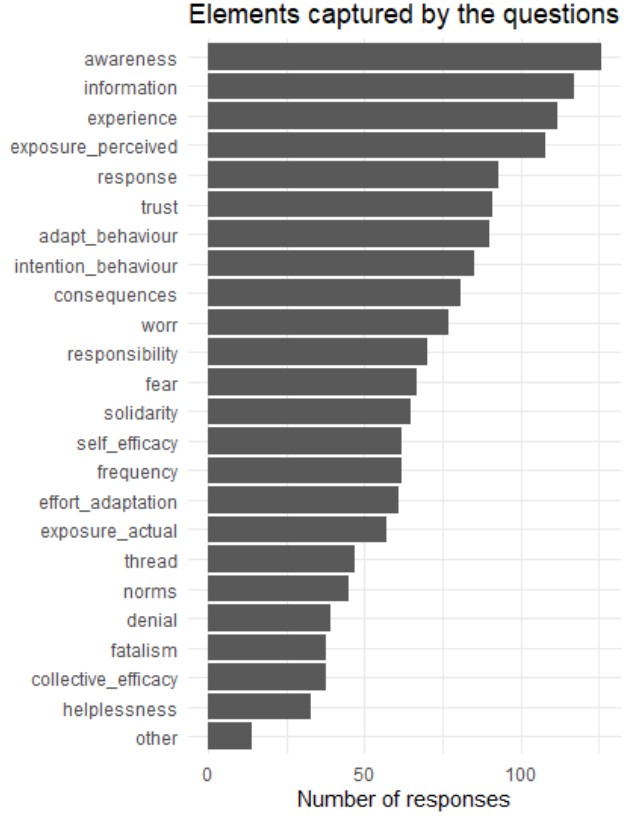

**Figure 4: Responses to** *'What did you try to capture with your risk perception questions?'* **The respondents could select multiple options.**




The highest numbers of mentions were directed towards knowledge-related elements such as awareness, information or experience, which each received up to 120 responses per element (80% of respondents). The lowest numbers, but still up to 50 responses (33% of respondents), were given for 'helplessness', 'collective efficacy', 'fatalism' and 'denial, wishful

thinking'. Around 20 (13%) respondents selected the option 'other'. Low numbers of responses should not be over-interpreted: they may be as important but there is less agreement on their relevance or awareness of their use amongst the respondents. Actual exposure was much less mentioned than perceived exposure. Of course, there may be biases introduced, as some elements could be considered similar. For example, the combined number of responses for 'fear' and 'worry' exceed the highest numbers per element recorded. However, the correlation between the two is only 0.55, indicating that the

respondents make a difference between them. While 55 (37%) respondents said they used both 'worry' and 'fear' in their studies, 22 (15%) used 'worry' but did not use 'fear', 12 (8%) only used 'fear', and 61 (41%) used neither. We therefore encourage reading the results carefully and want to leave interpretation as open as possible to foster discussion.

| | As 1st | As 2nd | As 3rd | Sum |
|---|---|---|---|---|
| Do not know | 49 | 50 | 54 | 153 |
| Awareness | 24 | 7 | 3 | 34 |
| Information, knowledge | 13 | 14 | 6 | 33 |
| Response, coping | 1 | 7 | 20 | 28 |
| Previous experience | 13 | 8 | 5 | 26 |
| Adaptive behaviour (actual) | 7 | 5 | 13 | 25 |
| Exposure (perceived) | 8 | 9 | 2 | 19 |
| Trust | 7 | 5 | 4 | 16 |
| Consequences, severity, impacts | 4 | 6 | 3 | 13 |
| Solidarity, social support | 3 | 5 | 5 | 13 |
| Exposure (actual) | 3 | 4 | 5 | 12 |
| Responsability | 2 | 5 | 2 | 9 |
| Fear | 2 | 2 | 4 | 8 |
| Frequency, probability | 2 | 3 | 3 | 8 |
| Worry | 3 | 2 | 2 | 7 |
| Efficacy (self) | 2 | 3 | 2 | 7 |
| Collective efficacy | 3 | 0 | 3 | 6 |
| Threat appraisal | 1 | 1 | 3 | 5 |
| Norms, injunctions | 1 | 3 | 1 | 5 |

**Figure 5: Responses to *'From your experience, what would be the three decisive questions or themes for cross-study comparability?'***
**Only items with 5 or more responses are shown.**

Regarding comparability, around one third of respondents chose to skip the question, *'From your experience, what would be the three decisive questions or themes for cross-study comparability?' or* stated that they did not know (*n* = 49, 32% for the first item, 33% for the second, and 36% for the third). In contrast to the convergence on the most used questions (Fig. 4), there was a wide dispersion on the most decisive questions or themes for cross-study comparability (Fig. 5). Adding the

three possible answers, the most cited items were 'awareness' (*n* = 34, 8%), the first choice for 16% of the respondents; then





‘information, knowledge’ (*n* = 33, 7%); ‘response, coping’, often a third choice (*n* = 28, 6%); ‘previous experience’ (*n* = 26, 6%); and ‘adaptive behaviour’ (*n* = 25, 5%), whereas the others were mentioned less than 5% of the time.

There was more agreement regarding research design choices (Fig. 6). The design of interviews, questionnaires or focus groups most often relied on the literature (*n* = 115, 77%), ‘discussion with co-authors’ (*n* = 96, 64%), ‘previous (own)

studies’ (*n* = 84, 56%) and ‘on my own’ (*n* = 71, 47%). Fewer respondents considered ‘other studies to compare’ (*n* = 78, 52%) or ‘discussions with practitioners or decision-makers’ (*n* = 56, 37%). This pattern may constrain the comparability of studies, a key interest of our Risk-SoS study – to find out how studies can be better compared or designed to be comparable. Almost one quarter of the respondents designed studies without considering previous studies or ‘the literature’. While the relatively large share of studies designed ‘on my own’ (47%) might recall the share of studies not using theoretical models or

frameworks (35%), there was no significant relationship between them – in fact, not relying on theory had no significant relationship with any of the answers on design choices.

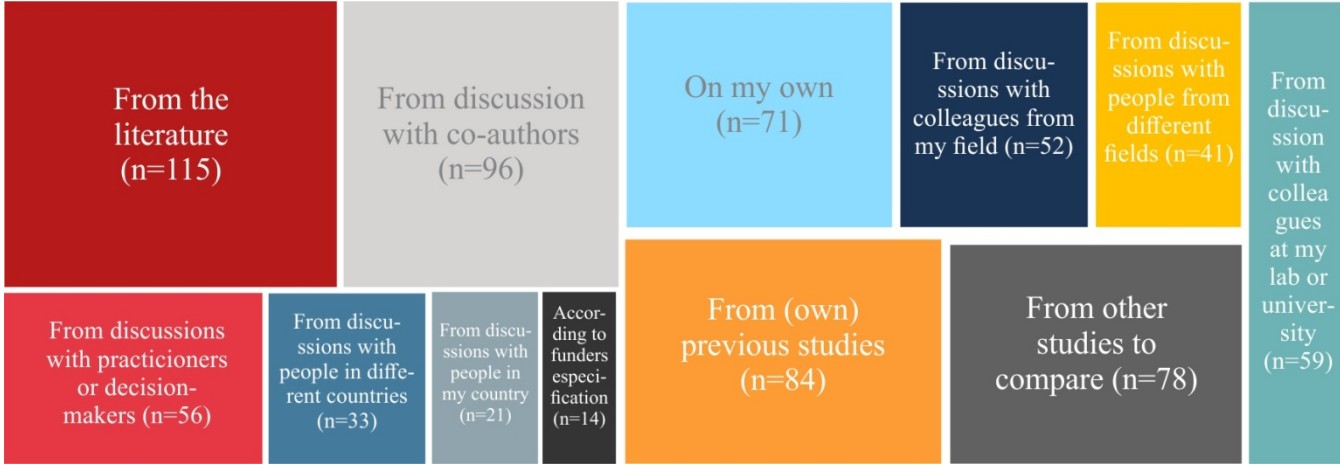

**Figure 6: Responses to** *‘How are your interviews, focus groups or questionnaires usually designed?’*

The same was true for how the questions were selected (Fig. 6). Respondents had most often used literature review (*n* = 128,

85%), ‘previous (own) studies’ (*n* = 79, 53%), ‘comparison with other studies’ (*n* = 74, 49%), ‘exploratory approach’ (*n* = 65, 43%) and ‘experience and habits’ (*n* = 60, 40%) to identify key questions. Therefore, the convergence on awareness, knowledge, experience and exposure questions might reflect the agreement on reliance on past choices – either by relying on the literature or habits. It did not, however, result in the recognition of the relevance of these choices for improving the field. A shortcoming pointed out by a participant during our webinars is that we did not offer respondents the option to declare that

they based their questions and methods choices on theory and only on the literature, previous studies, among others.





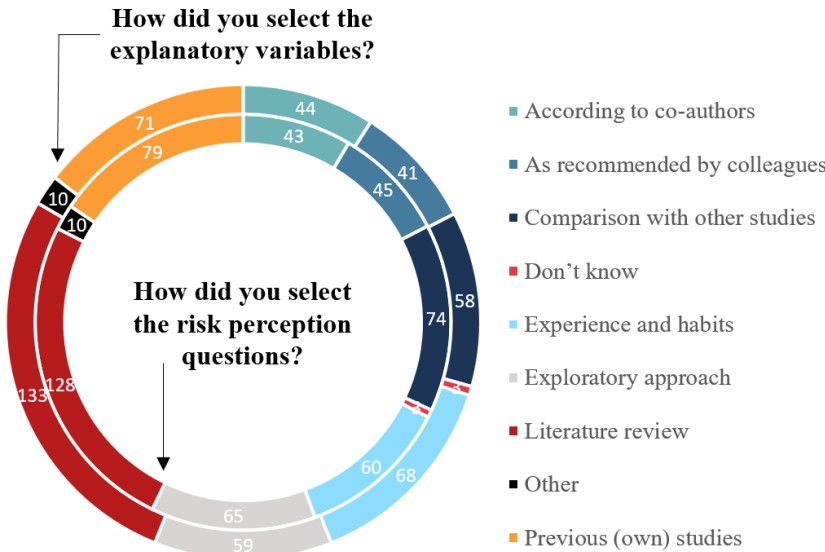

**Figure 7: Responses to** *'How did you select the risk perception questions?'* **(inner circle), and** *'How did you select the explanatory variables?'* **(outer circle).**

Overall, answers on the selection of explanatory variables were consistent with those of other design questions (Fig. 6). Most respondents selected them by considering literature review ($n = 133$, 87%), leaving more than one in ten failing to consider previous studies for their design choices. Around half of the respondents based the selection on 'previous (own) studies' ($n = 74$, 49%) or 'experience and habits' ($n = 68$, 45%). Again, only a minority considered 'comparison with other studies' relevant for their design choices ($n = 58$, 39%). Among the respondents declaring that they did not rely on theoretical models or frameworks ($n = 53$), 80% subsequently declared that they did base their variable selection on the literature ($\chi^2 = 5.86$, p = .01). The other variable selection choices had no significant relationship to the answers on theory. It might be argued that while the respondents did not rely on specific theoretical models themselves, they indirectly incorporated the theoretical framing from previous studies in their own design choices. However, the tendency to reproduce past research design choices and the dissatisfaction with them, or the lack of convergence on choices that might improve cross-study comparability, points in the opposite direction. Therefore, a closer look at the selection of explanatory variables and the drivers of these choices is required.

# 5 Explanatory variables: panic in the multiverse?

Another interest of this study was to identify common explanatory elements and variables used to explain risk perception and adaptive behaviour. Previous studies have shown that results depend on the input of variables (Lechowska, 2018), and a model does not necessarily improve with a greater number of variables (Rufat et al., 2020). In our study, the sheer variety of explanatory variables in use and the divergence in research design choices might give the impression that studies run in





245 parallel universes. As a result, studies of risk perceptions and behaviours might appear more like a multiverse than a consolidated academic field. Yet, panic seems unnecessary, as this situation might be the momentary price that is paid without further reflection in research about an ongoing, uncoordinated, exploratory multidisciplinary effort.

Socio-demographic characteristics are the most contested drivers of risk perception. This led us to ask respondents not only to mention the explanatory variables they have used to study risk perception and behaviour but also to identify the three most

250 relevant variables for cross-study comparability and long-term monitoring (Fig. 8). The surveyors reported having applied a wide diversity of variables to explain variation in risk perceptions and behaviours. Unsurprisingly, socio-demographic characteristics (age, gender, education, income, family or household composition, and occupation) were the most often used. Certain risk or environmental factors were also mentioned frequently, most notably previous hazard experience and hazard exposure. 'Age', 'gender', 'education' and 'previous hazard experience' each received more than 100 responses (67%). In

255 addition, over one third of the respondents chose a few external or contextual factors (vulnerability or resilience), whereas other personal factors were much less commonly used (minorities, disability or language proficiency), and informational factors were largely absent. A moderate number of respondents had used health, insurance demand, anxiety or resilience and its determinants, that is, social capital and coping capacity. Factors of social vulnerability such as language proficiency and whether the surveyed individual had a disability or was a minority were rarely applied. These results are consistent with

260 literature reviews published in recent years (Moreira et al., 2021; Rufat et al., 2020; Lechowska, 2018; Renn et al., 2000).

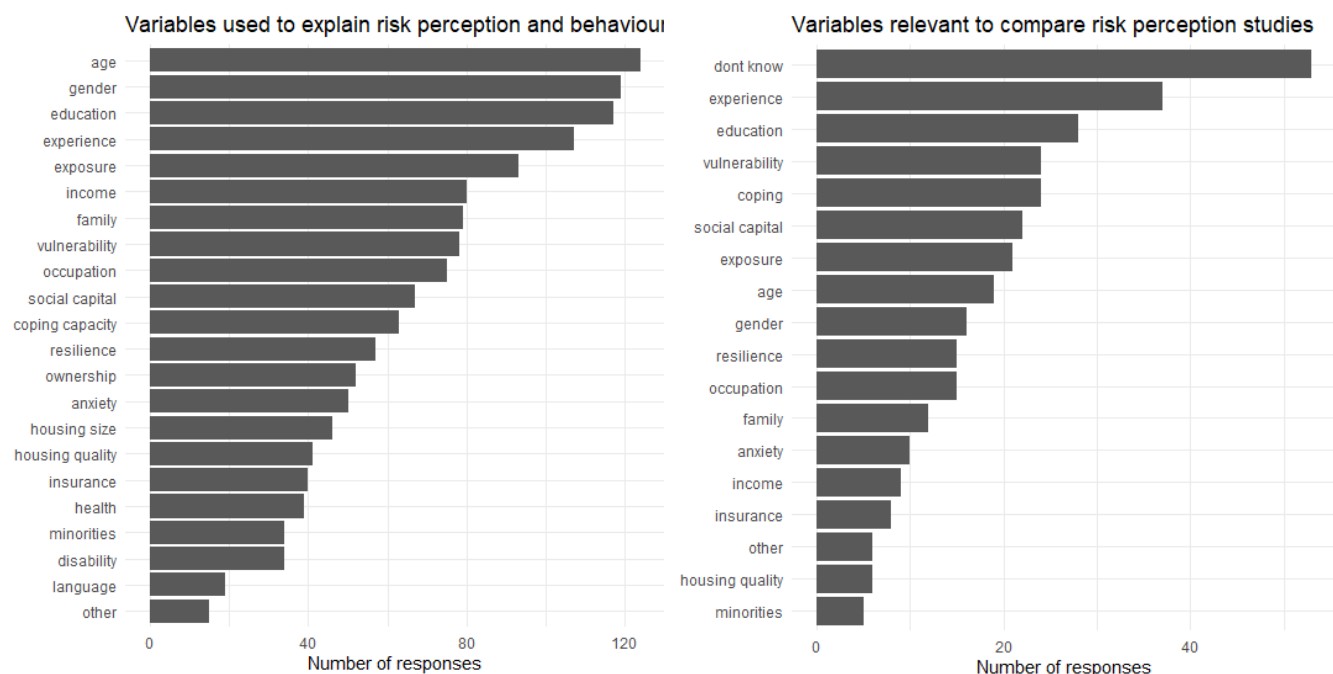

**Figure 8: Most used variables to explain risk perception and behaviour (a, left) and variables (three choices) identified as most relevant for cross-study comparability and long-term monitoring (b, right).**





The three most often mentioned variables were ubiquitous – found in most databases – and matched general demographic characteristics. The fourth and fifth most selected variables, 'previous hazard experience' and 'exposure to hazards', were connected
to the context of risk. Both reached similar high rankings in the questions asked by surveyors (Fig. 5), likely because they cannot be derived from standard databases and therefore must be collected by surveyors. It is worth noting that 'vulnerability' was mentioned more often than 'resilience' to explain risk perception and behaviour (Fig. 8a), which may be linked to the theoretical frameworks used to design the studies. Of the 21 options, 'health', 'minorities', 'disability' and 'language proficiency' were each mentioned by less than one quarter of the respondents. Fourteen (9%) respondents indicated that there were 'other' useful
variables that were not included in our survey. This may point towards a need for further investigation.

What stands out is the discrepancy between the variables used (Fig. 8a) and the variables thought to be critical to cross-study comparability or long-term monitoring (Fig. 8b). Around half of the respondents declared that they did not know which variables were useful to ensure comparability. This result might reflect the current disagreement on risk perception drivers and challenges in directly comparing the current collection of independent case studies. While socio-demographic
characteristics (age, gender and education) were often used, followed by risk or environmental factors (experience and exposure) and contextual factors (vulnerability and resilience), the ranking is reversed in the case of comparability or long-term monitoring: environmental factors come first, followed by contextual factors, and socio-demographic factors are less mentioned, except for the role of education.

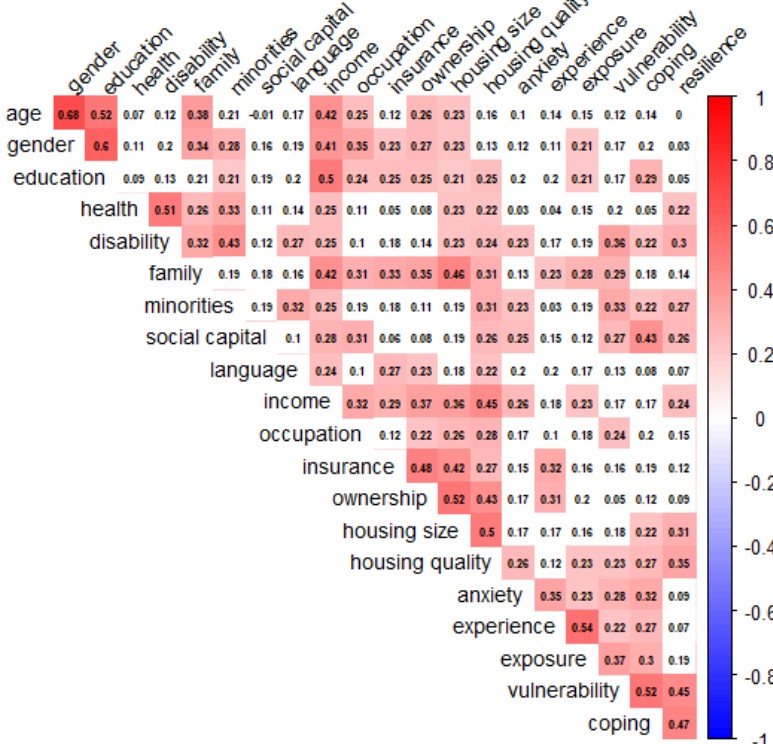

**Figure 9: Heatmap of the correlations among the variables used to explain risk perception and behaviour.**





Figure 9 illustrates a Pearson correlation matrix of all the variables, with statistically significant correlations (p < .01) highlighted in red for positive correlations. These correlations are as expected; they identify groups of variables often used together (e.g., age, gender and education), and the least frequently reported have fewer relationships, even though environmental factors (experience and exposure) have fewer relationships than personal factors.

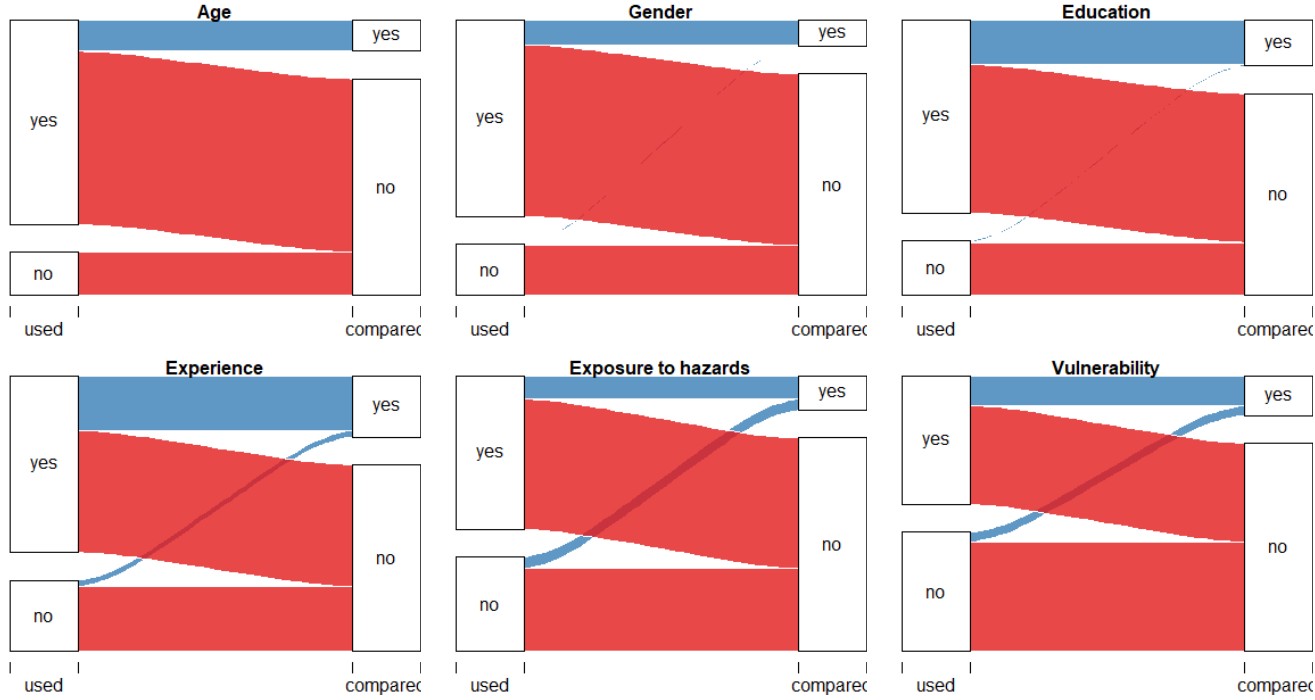

**Figure 10: Explanatory variables used vs mentioned as important for cross-study comparison. For instance, 124 respondents used age as an explanatory variable, but only 19 of them (15%) thought age was relevant for cross-study comparison.**

Comparing the explanatory variables used and declared relevant for comparison by the same respondents (Fig. 10) reveals two contrasting situations. While most respondents used socio-demographic characteristics in their studies, a minority of them considered such factors important for comparison, whereas virtually none of those who did not use them considered them important for comparison. Conversely, while a smaller proportion of the sample used environmental factors (experience and exposure) and less than half used contextual factors (vulnerability and resilience), a substantial proportion of those who did not use them did nevertheless consider that they might be important for comparison, whereas a larger share of those who did use them declared them critical for comparison. There is, however, no agreement, as no single driver was mentioned by more than one quarter of respondents as one of the three most important for case study comparison and monitoring.

This leaves us with the challenge of fostering convergence among the wide diversity of risk perception and behaviour drivers, as there was no agreement on their relevance for comparison or long-term monitoring. This might explain why most respondents declared that they based their variable choices on the literature and their own previous studies (Fig. 6) – there does not seem to be any other robust criterion at the moment.



**Figure 11: Explanatory variables selection according to the background of respondents and study characteristics**

The ranking of variables analysed according to the hazards studied, the location diversity of case studies, the disciplinary background and study size (Fig. 11). The (maximal) sample size may have a strong effect on the distribution of the explanatory variables: Respondents using smaller samples had used environmental (experience and exposure) and contextual factors (vulnerability and resilience) more often, whereas respondents using larger samples had used income, ownership and anxiety. The ranking was only marginally impacted by the hazards studied, with insurance, ownership and home

characteristics slightly more used for floods; experience, vulnerability and anxiety for multi-hazard studies; education and vulnerability for geophysical hazards; and resilience and health for epidemics. The location diversity of the case studies – only in one country or in several – affected the drivers used. Respondents with more case studies diversity used gender, coping capacity or minorities more often, whereas respondents with less diversity were more likely to use age, exposure,





occupation, housing size or location. The disciplinary background of the respondents (main field of PhD or studies) had
almost no impact, which may be linked to the interdisciplinary focus of most studies.

The respondent's experience, seniority, and methods used had a lesser impact on the ranking of explanatory variables. The respondents' experience (number of case studies conducted) had little effect on the drivers used. Respondents with more than five case studies may have used minorities, language proficiency, and family or household size or composition more often. In contrast, respondents conducting their first study may have used age and gender less often and contextual factors more
often. The respondent's seniority (years since PhD) also had negligible impact on the drivers used, even though contextual factors were used by junior investigators more often, whereas more senior surveyors used education, occupation and livelihood. While respondents exclusively using interviews or focus groups were more likely to use contextual factors and less likely to use income, insurance or ownership factors, the impact of methods on the ranking of drivers was weak.

We also tested for the gender – half the respondents were female – and the institutional affiliation of respondents, inside or
outside academia: These characteristics did not affect the ranking of explanatory variables.

# 6 Case studies and regional patterns

Contrary to our expectations, the Risk-SoS survey did not capture strong regional differences in risk perception and behaviour approaches. The location of the case studies conducted by respondents – only in Europe, only outside of Europe or a combination of both – had little effect on the ranking of explanatory variables (Fig. 12). Respondents in our sample working on
non-European case studies less often used age, experience or income and more often used gender, exposure, vulnerability or occupation. Respondents combining case studies more often used education, family or household size, or coping capacity. Similarly, the work environment of the respondent – inside or outside of Europe – had little impact on the risk perception and behaviour drivers used, although respondents outside of Europe used education, ownership or insurance less often than gender, occupation and livelihood, or health.

Although the sample size is not appropriate to use inferential statistics due to high data scattering, we tested for hypotheses of regional difference in risk perception and behaviour approaches and assessments, on the one hand, between respondents based in or outside Europe and, on the other hand, between different regions of Europe. Most of the time, we did not find a statistically significant link (p = .01) with the theories in use or the selection of the explanatory variables. The only time a chi-square test found a statistically significant link, logistic regression rejected the association. The same result was obtained
when testing for a dichotomous division between Eastern and Western Europe regions, despite the barriers in scientific communication during communism and most of the early post-communist transition period.





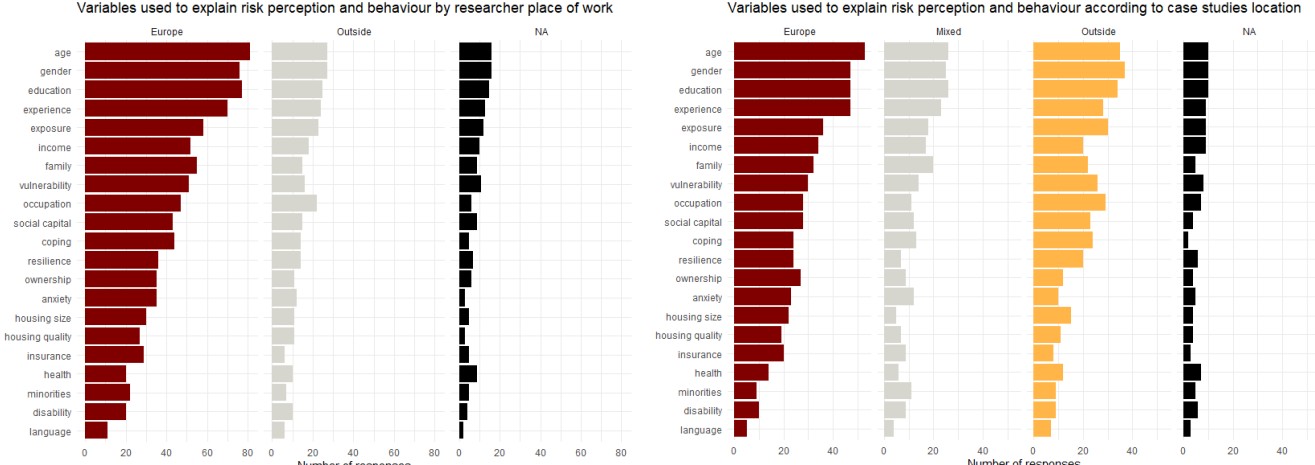

**Figure 12: Ranking of explanatory variables according to location**

Even though our study did not capture significant regional differences in research design and explanatory variables selection, we can state that the short-term horizons used by households were more frequent in context-specific hazard research in the former communist states of Eastern Europe (Raška, 2015). However, this observation is blurred by the imbalanced background of the participants in our sample and the prevalence of geographers and researchers from environmental sciences with a focus on inductive hazard-related approaches, indirectly informed by specific risk and vulnerability theories. Another regional discrepancy in our results that may be explicable from a historical perspective was that most researchers in former communist countries were limited to studies inside their country. At the same time, researchers in Western Europe remain more open to case studies located in other countries, even on other continents. It is however appropriate to underline that the initial focus of the Risk-SoS survey was on surveyors based in Europe.

# 7 Impact of COVID-19 on research

As the Risk-SoS survey was disseminated from December 2020 to April 2021, we inquired about the impact of COVID-19 on respondents' research. Contrary to our expectations, only 53% of respondents ($n = 80$) declared an impact on their research on risk perception or behaviour (Table 2). It is worth noting that 10% did not know ($n = 15$), which leaves a little more than a third of respondents declaring no impact ($n = 55$, 37%).

| Has Covid-19 impacted your own research? | n | % |
|---|---|---|
| Yes | 80 | 53 |
| No | 55 | 37 |
| Don't know | 15 | 10 |

**Table 2: Responses to 'Has Covid-19 impacted your own research on risk perception and/or behaviour?'**

An optional open question was offered to respondents to elaborate on their answer if they wished. We collected 69 different open answers (representing 46% of respondents). Unsurprisingly, the impacts reported were mostly negative, such as





impairments in access to people, colleagues, travel, fieldwork and traditional methods demanding face-to-face conversations. An increased workload and work-life balance issues were also mentioned several times, as well as reluctance of surveyed people, postponed empirical studies and an inability to conduct planned follow-up surveys. However, the adaptation to online methods, either immediate or planned for future surveys, points towards an adjustment out of necessity rather than resignation or cancellation of all empirical work during the pandemic. Some respondents mentioned that online interviews or 360 surveys had costs and practical benefits after adjusting to the new methodologies, while a few acknowledged that online meetings and webinars facilitate exchanges with a scattered research community.

# 8 Discussion

## 8.1 Discussion of the results in relation to the existing literature

Our finding that most risk perception studies are not strongly embedded in a theoretical framework is consistent with review
studies of risk perception research for particular types of hazards. For instance, Kellens et al. (2013) review 57 peer-reviewed articles on flood risk perception and communication and conclude that most studies are exploratory in nature and not based on a theoretical framework. The exploratory nature of flood risk perception research was also observed in an updated review of this research field by Lechowska (2018). Our observation that PMT is the most commonly applied theory is consistent with the review of flood risk mitigation behaviour by Bubeck et al. (2012), who point towards its relevance in
explaining behaviour. A meta-analysis of determinants of climate adaptation behaviour by van Valkengoed and Steg (2019) also concludes that PMT variables are strong predictors of this behaviour, and hence a suitable theoretical framework in this particular strand of research. Lechowska (2018) concludes that the main flood risk perception indicators used in the literature are awareness and worry. This is consistent with our findings for the broader risk perception literature, if we consider fear to be similar to worry and combine these two variables in one category. However, our results show that surveyors distinguish
between them.

Regarding explanatory variables, Kellens et al. (2013) conclude that almost any study on flood risk perception includes socio-demographic variables. They also point towards the importance of previous flood experience as an explanatory variable that is commonly used to test the availability heuristic (Tversky and Kahneman, 1974) – that is, whether people perceive risk to be higher if they can easily recall an occurrence of the event, because they experienced it in the past. van
Valkengoed and Steg (2019) also observe that disaster experience is commonly used as an explanatory variable in the literature on climate adaptation behaviour. However, socio-demographic characteristics are the most contested drivers of risk perception. For example, some studies observed that people with less education worried more about risk (Bradford et al. 2012), while others found no such effect (Kuhlicke et al., 2011), and some attributed such an effect to the relationship between education and income (Wachinger et al., 2013). Similarly, some studies conclude that immigrants and socially
vulnerable communities have lower levels of self-protection and higher risk perceptions (Armas, 2008), whereas others





attributed such effects to other characteristics, mostly age and income (Adelekan et al. 2016) or residential segregation (Rufat, 2015). Some studies also claim that older and higher-income residents have higher risk perceptions and more often adopt precautionary measures (Grothmann et al., 2006), whereas other studies find that age (Armas et al., 2015; Botzen et al., 2012) or income (Lindell et al., 2008; Botzen et al., 2009) have no significant impacts. Such contradictory evidence on

behaviour hampers recommendations for policy and risk management (Lechowska, 2018), such as the design of targeted risk communication strategies (Höppner et al., 2012). As many studies focus on different dimensions (sociological, economical or psychological), internal or personal factors (gender, age, education, income, values or trust), external or contextual factors (vulnerability, institutions, power, oppression or cultural backgrounds), risk or environmental factors (perceived likelihood or experienced frequency) or informational factors (media coverage, experts or risk management), their diverging sets of

variables, methods and approaches are scarcely compatible. Our results reflect this diversity of methods, theories, questions and explanations, as well as the discrepancy between the variables used and the variables thought to be critical to cross-study comparability or long-term monitoring.

## 8.2 Towards a list of minimal requirements to compare studies (Goal 1)

Our results map the diversity of practices and present shortcomings in the field. While they signal that cross-study comparison

is not the primary concern of surveyors when they design their research, they offer two possible ways forward to improve convergence, comparability and cumulative knowledge. One is factual and relies on what surveyors are currently doing. The other is counter-factual and relies on what surveyors may have more carefully considered to ensure their study comparability. We consider six questions critical for ensuring comparability during the research design phase of a study, as follows:

1. Is there a set of explicit hypotheses specified?

2. Are the hypotheses formally derived from one or more theories?

3. Are the constructs (e.g., risk awareness or trust) derived from one or more theoretical frameworks?

4. Are there research questions or themes that are comparable with those of previous studies?

5. Are there explanatory variables derived from previous studies?

6. Do the results allow for a formal test of the hypotheses or theories while controlling for context and other variables?

Answering all six questions positively would ensure that the designed study is likely to lead to comparable results, if other studies have applied the same theoretical framework (or parts of it). In general, building a larger theory-informed empirical evidence base may facilitate meta-analyses and allow for the systematic identification of context-dependent effects. Moreover, producing cumulative knowledge in this way may assist policymakers in grounding their decisions in plausible and coherent mechanisms of action. On the other hand, 'forgoing theories may result in measuring a wide range of less

relevant, marginally relevant, or irrelevant constructs, while also minimising the chances of obtaining results that are meaningful and not by pure chance' (Bhattacherjee, 2012: 21).



However, such an approach might prove to be a major deviation from current practice according to our Risk-SoS results. An easier but less satisfactory solution – which may be only transitory – would be to follow the revealed trend in the field to base design choices on previous research. In other words, a more tangible way forward would be, as a first step, to implement some of the currently most-in-use questions, themes, constructs and variables – i.e. the top ranking in our results – in future studies, without expending effort to improve the theoretical foundation and methodological robustness of the research design. While we acknowledge that the ranking produced may not be definitive, this type of instrument may be a critical way forward to bridge the current research design gap in the field but, to the best of our knowledge, is missing. We do not promote a single umbrella theory, unique standardised method or some one-size-fits-all global questionnaire. Nevertheless, reproducing (at least some of) the currently most frequently used questions and explanatory variables in future case studies may be the most favourable way forward.

## 8.3 Can we reach shared criteria to address context-specific aspects? (Goal 2)

Our Risk-SoS survey did not reveal significant regional differences in risk perception and adaptive behaviour study design. The reasons are manifold, including our initial focus on researchers based in Europe. However, as risk perception, behaviour and adaptation are locally embedded practices and social, institutional and cultural factors play a key role in driving or hindering adaptation behaviour (Berrang-Ford et al., 2021), more comparative research is necessary. The issue is – for example – when a study in Italy says that gender has an effect on perceptions while one in Romania says that it is not gender but age. At the moment, we cannot investigate if this is related to the context (country) or to the methodological choices of the study (question, theories, etc). Therefore, relying on a unified theoretical framework and following the procedure outlined in the previous section seems particularly relevant for such a cross-regional comparison; then, more context-specific drivers can be identified. However, it is still an open question at which spatial level such comparative studies should be conducted (e.g., continental, country, local level). We suggest that any kind of comparative study is highly relevant as there are so few. This specific issue is being collectively discussed in the Risk-SoS webinar to help disentangle the various effects (contextual, methodological and casual).

## 8.4 Improving comparability and long-term monitoring (Goal 3)

The elements most often captured by respondents were, in descending order: risk awareness, information (knowledge), previous hazard experience, perceived hazard exposure and coping with disaster (response). The ranking is similar for those considered decisive for cross-study comparability – if we set aside the fact that 'don't know' was by far the most frequent answer – with the addition of another item: adaptive behaviour (actual, not projected). The most often used explanations or variables were, in descending order: age, gender, education, previous hazard experience, actual exposure and income. The respondents made a different ranking, however, for the explanations they consider the most important: previous hazard experience, education, age, gender, vulnerability, coping capacity and social capital. The rankings were more scattered for





those considered decisive for cross-study comparability – again with 'don't know' by far the most frequent answer – but were more likely to include previous experience, vulnerability, coping capacity and social capital than age or gender. Thus,

ensuring that future research designs consider collecting all these themes and control for all these explanations should be considered good practice in the field. However, large shares of replies were 'don't know', and the fact that surveyors use themes and variables does not qualify them as decisive for comparison. Such a discrepancy between use and reputation is a reminder that neither of those guarantees merit. This might explain why most respondents said that they base their choices on their own previous studies – it remains hard to find other robust criteria at the moment.

Unfortunately, our study points to specific challenges for comparability and long-term monitoring. Even highly experienced researchers – with over 20 completed studies or over 20 years of research experience – struggled to narrow down the core questions of risk perception and behaviour, reduce complexity to a few key themes and variables or agree on the most significant ones for comparison and long-term monitoring. As a substantial share of studies failed to rely on the literature, previous studies or theories and frameworks to strengthen their research design, challenges in comparing results are

expected. While most respondents used risk perception as an explanation of behaviour and adaptation ($n = 97$, 65%), the majority of studies were observational ($n = 120$, 80%), and just over one third had implemented their studies or surveys multiple times ($n = 62$, 41%). The dispersion of studies combined with these choices do not favour causality detection, the assessment of intervention effects, nor sequential disasters cumulative effects, nor drawing robust lessons to guide policy and help risk communication strategies. The disuse of common theoretical frameworks may add to this problem. Only one third

of surveyors conducted formal tests of the validity of a theory or the power of an explanation ($n = 50$, 33%) and only half of those that did formally compared two or more theories in the same study ($n = 25$, 17%). Without a substantial and enduring convergence effort, comparing the merits of different theories, assessing the worthiness of different explanations, not to mention long-term monitoring, can only be achieved by studies designed by the same team or inside a group of like-minded surveyors. Beyond the issue of reduced efficiency and speed, this places context specificity assessment, temporal and

analysis scale variation or cross-validation out of reach.

In other words, we recommend that future studies implement all items listed above, along with their specific questions, and test for a wider set of explanations or demonstrate which of them lack merit for their specific case study or context – and present this explicitly as a result – before discarding them from their research design. However, our study does not intend to promote a single theoretical framework or make assumptions about why many seem not to use such frameworks. There may be good reas-

ons to avoid using pre-existing frameworks or to use an inductive approach, especially because understanding risk perception, or the bridge between perception and adaptive behaviour, has evaded most explanatory frameworks or models so far.

# 9 Conclusion

This study initiated a discussion on standards on risk perception, behaviour and adaptation research. Although we reached many surveyors ($N = 150$), our empirical basis has sampling constraints. The results point towards further research and





discussion aiming to inform the community about key findings and persisting gaps. While using theoretical constructs allows comparing and accumulating evidence, most of these studies are exploratory in nature. Over one third of surveyors did not rely on a particular theoretical model or framework to guide their studies. Only one third of surveyors tested the validity of a theory or the power of an explanation. Even fewer formally compared two or more theories in the same study. These limitations might be the momentary price to pay for an ongoing multidisciplinary effort. However, the exploratory and

fragmented nature of current studies may make them look like fishing expeditions, finding results mostly by chance and reaching conclusions that other studies cannot substantiate.

While the diversity of approaches is an asset, the robustness of methods is an investment. Surveyors reported a tendency to reproduce past research design choices. They also expressed frustration with this trend, and one third of surveyors did not know how to improve the situation in the field. We recommend greater attention to the formalisation and robustness of

methods and advocate reaping the benefits of the current diversity of choices by systematically comparing different approaches. Similarly, we recommend that future studies test for a broader set of explanations or demonstrate which of them lack merit for a specific case study or context before discarding them from their research design.

The wide diversity of opinions on how to remedy comparability is a cause for concern: convergence and comparison remain high-hanging fruits in the field. The discrepancy between actual usage, estimated utility and belief in the merit of cross-

validation or long-term monitoring of the current wide range of potential explanations is equally worrying. One way forward is counter-factual and relies on what surveyors should more carefully consider, especially (1) grounding their research design in theory; (2) improving the formalisation of methods; and (3) formally comparing theories, methods and explanations. Another is factual and relies on what surveyors are currently doing. It involves (1) ensuring that future research designs consider collecting all the most-in-use themes and (2) controlling for all the most-in-use explanations, if they can be

theoretically linked to the research questions and logically implemented in their models and methods.

**Author contribution**: SR, AF and CK originally conceived the study. SR, AF, EC, PR, IA, WB, and CK designed the survey with inputs from the community during the Risk-SoS webinars. SR did the data collection and curation. SR, MMdB and IA analysed the results of the survey, and all authors contributed to the interpretation of the results with further inputs from the participants to the Risk-SoS webinars. SR and MMdB wrote the first draft of the paper, to which all authors contributed.

**Competing interests**: The authors declare that they have no conflict of interest.

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
