# Peer review of "Surveying the Surveyors to Address Risk Perception and Adaptive Behaviour Cross-study Comparability"

_Natural Hazards and Earth System Sciences, 2021_

## Referee Comment (RC1)

Manuscript Title:    Surveying the Surveyors to Address Risk Perception and Adaptive Behaviour Cross-study Comparability

*Summary:* This manuscript describes a uniquely intriguing and informative examination of research practices in the field of risk perception and protective action. The authors have addressed some important issues that are related to, but distinctly different from, those associated with the replication crisis (e.g., Shrout & Rodgers, 2018). The 150 member sample is sufficiently large to draw some reasonable conclusions even though it has a very high proportion of European researchers studying floods. In the absence of a census of researchers and hazards in the field of risk perception and adaptive behavior, it is unknown whether these sample characteristics are representative of the field's researchers. Nonetheless, the authors have identified and summarized problems that I have seen repeatedly during my 50 years studying floods, volcanic eruptions, hazardous materials releases, earthquakes, hurricanes, tornadoes, and tsunamis in the US and other Pacific Basin countries. Consequently, their data are likely to generalize to other hazards and countries. In particular, their data on theoretical frameworks, hazards studied, research designs, and sources of explanatory variables for risk perception and protective action provide ample support for their six recommendations for improving comparability of research results. I strongly endorse publishing this manuscript, but also have some suggestions for improving it.

| Line | Comment |
|------|---------|
| 52 | It is important to distinguish between inconsistent findings and contradictory findings. Inconsistent findings occur when a variable that has statistically significant correlation/regression coefficients in some studies but nonsignificant coefficients in other studies, whereas contradictory findings occur, for example, when a variable has coefficients of opposite signs in different studies. These problems can be addressed by statistical meta analyses showing that the average effect size is not significantly different from zero (for inconsistent findings) or that a moderator variable can account for the differences in the direction of effects (for contradictory findings).Of course, concluding that research findings are inconsistent does not explain *why* those findings are inconsistent but, as noted below, inconsistent findings about psychological variables are likely due to inconsistent operationalizations (i.e., indicator variables) of latent variables. |
| 62 | There are other reasons for encouraging individual households to become more resilient, such as the large scope of destruction in a disaster preventing community emergency managers from providing immediate assistance. For example, emergency managers in some US Pacific Coast jurisdictions advise households that they must be self-sufficient for up to 14 days after a major earthquake because destruction of transportation infrastructure will prevent the authorities from delivering assistance. |
| 73 | It would be helpful to be more specific about what is meant by "opposite conclusions". Does this mean A does vs. doesn't have a significant relationship with B, A has a positive vs. negative relationship with B, or A causes B vs. B causes A? |
| 87 | The term "minimal requirements" could benefit from greater specificity. Does this refer to theoretical grounding, population sampling, psychological measurement (i.e., the reliability and construct validity of the latent variables), statistical analysis (e.g., compliance with statistical model assumptions), or other aspects of the study design? |
| 184 | I was initially surprised that *Response* was only fifth on the list, but I noticed that *Adaptive behavior* was listed seventh, and *Intention/behavior* was listed eighth, so it is clear that behavioral issues are more important than they might seem from cursory examination of Figure 4. Further examination of this figure showed that other topics could also be grouped—for example, *Worry, Fear, Denial, Fatalism,* and *Helplessness* as affective responses. However, that categorization |

would depend on some important theoretical distinctions, such as automatic vs. controlled processes (e.g., Moors & De Houwer, 2007), where *Worry, Fear,* and *Helplessness* might be considered examples of automatic processes, but *Denial* and *Fatalism* might be considered examples of controlled processes. It would be interesting to see these elements grouped according to their hierarchical clustering.

203    As with Figure 4, it seemed surprising initially that *response/coping* was only nominated once as the first item and is third on the list of summed nominations. However, if *Response/coping* is combined with *Adaptive behavior*, the combined category would be first on the list of specific summed nominations with 53 mentions.

226    Figure 7 would be easier to understand if *Comparison with other studies* and *Literature review*, which seem synonymous, were adjacent. Similarly, *Experience and habits* and *Previous (own) studies* seem quite similar, so placing them adjacent to each other would allow the reader to better understand their total effect. Of course, if each pair of categories is exactly synonymous, then summing them is double counting. Nonetheless, it seems clear that reviewing the published literature has the greatest effect, followed by continuity of the researcher's previous work, peers' recommendations, and finally, exploratory innovations.

239    I am unsure how to interpret the statement that "only a minority considered 'comparison with other studies' relevant for their design choices", given that the overwhelming majority (87%) reported relying on literature reviews.

240    The use of the term "panic" here might seem like a relatively innocuous example of the hyperbolic use of that term, but I find it concerning because it tends to perpetuate a myth that disaster researchers have long debunked (Fritz & Marks, 1954). The problem with using the label "panic" as anything other than a clinical diagnosis by a qualified mental health professional is its misuse by the news media and misinterpretation by public authorities (Clarke & Chess, 2008). In the present manuscript, the phrase "panic seems unnecessary" could be replaced quite satisfactorily by "there is no need for undue concern".

247    Given the data in Figures 6 and 7, which show that research designs are heavily influenced by comparison with other studies and literature reviews, it seems inappropriate to characterize risk perception research as an "uncoordinated, exploratory" effort. A more appropriate term might be an "organized anarchy" (Cohen et al., 1972) of very loosely coordinated research efforts. I have found that the most experienced researchers (more or less defined by those with the most publications in the field), and less experienced researchers who have studied with the most experienced researchers, operate within self-defined domains that are *coordinated implicitly* (by recognizing the limits of their own research paradigms) and *substantially confirmatory* (assessing the generalizability of their paradigms by applying it to different hazards and populations). Instead, "uncoordinated, exploratory" research appears to be conducted mostly by the other tail of the experience distribution—people who are drawn into the field by a highly publicized event and conduct a survey that is designed to test their (often incorrect) intuitions about the relevant variables without reviewing the existing literature. Many such researchers publish a single study in a journal whose reviewers are equally unfamiliar with the hazards/disasters literature and then move on to another topic. Their studies' weak designs and idiosyncratically measured constructs make it difficult to assess their contribution to the literature; they're not bad enough to dismiss entirely but not good enough to make a solid contribution.

252    The prevalence of age, gender, education, income, family or household composition, and occupation as explanatory variables seems relatively easy to understand. Despite evidence of their

weak and inconsistent correlations with behavioral variables such as evacuation (Baker, 1991; Huang et al., 2016), it is conventional to measure these variables so readers can assess the degree to which a survey sample is biased in comparison to census data for the geographical area in which the data were collected. Since these demographic variables have already been collected, it is a simple matter to examine their correlations with psychological and behavioral variables. Other possible reasons for the prevalence of these demographic variables are that their inclusion requires no knowledge of the research area and that they are compatible with any of the theoretical perspectives listed in Figure 2 (which are not necessarily compatible with each other).

272    The reversal of the demographic, environmental, and contextual variables between frequency of use and relevance to cross-study comparison is extremely intriguing. One possible explanation is an extension of my previous comment about the prevalence of demographic variables. Specifically, the frequency of use data can be explained by the fact that it takes no imagination to propose demographic variables as predictors of protective action and not much more imagination to propose environmental variables (exposure and experience) as antecedents of risk perception. However, the significance of contextual variables (vulnerability and resilience) has only been recognized more recently. The different rank ordering with respect to relevance could be explained by the fact that environmental variables such as hazard experience tend to be homogeneous within communities, especially for infrequent hazards, so studies focused upon a single community or small region will find little variation in hazard experience. Thus, it is necessary to survey communities that differ substantially in their hazard experience to obtain the requisite variation in experience at the household level (e.g., the Lindell & Prater, 2000, comparison of low-seismic Western Washington cities with high-seismic California cities). Since such designs are much more expensive than single community designs even though they are much more informative, that could explain why the difference between use and relevance.

280    The correlation matrix in Figure 9 is very useful, but it could be improved by conducting a hierarchical cluster analysis and then rearranging the variables according to the resulting clusters. For example, the current figure clearly indicates that age, gender and education form a cluster but closer examination suggests that other variables also belong in this cluster—definitely income and probably the housing variables.

300    The presentation of the results in Figure 11 would benefit from an explanation of the questions that were asked. For the upper left panel, were respondents asked in an open-ended question to identify the hazards they studied and then the responses were categorized as presented in the upper left panel? Were the data in the upper right panel generated by asking respondents to report the total number of communities they had studied or the maximum number of communities in the study with the largest number of communities? Were the data in the lower right panel generated by asking respondents to report the total number of respondents in all of their risk perception studies, the typical number in any of their risk perception studies, or the number of respondents in the study with the largest number of respondents?

317    It would be helpful to clarify the comparison group in the statement "While respondents exclusively using interviews or focus groups were more likely [than survey researchers?] to use contextual factors".

330    I don't understand the rationale for the statement "Although the sample size is not appropriate to use inferential statistics due to high data scattering". First, it appears from Figure 12 that there are enough cases to test the difference between European and non-European researchers in their use of age and many of the other variables at the top of the variable list. Of course, at the other extreme, it seems quite clear that there definitely are not enough cases to test such differences in

the use of language and many of the other variables at the bottom of the variable list. Similarly, it appears that there are enough cases to test the difference between studies in European, Mixed, and non-European locations on some, but not all, of the explanatory variables. Thus, there appears to be no basis for neglecting all statistical comparisons because of sample size.

Second, the authors seem to be using "high data scattering" to mean "large variance", but it is unclear if this means variance in counts across explanatory variables, across researcher locations/study locations, or both. Whatever the authors' intended meaning, it is unclear why this variation would be a problem. As noted in the paragraph above, there appears to be an adequate sample size for assessing differences on some, but not all, of the explanatory variables.

340    The statement "However, this observation is blurred … indirectly informed by specific risk and vulnerability theories" seems quite plausible, but the authors should provide the data analyses that support it or clearly acknowledge that it is a speculative explanation. The following statement "Another regional discrepancy…inside their country" does not have this problem because it is explicitly stated as a speculative explanation—"*may be explicable* from a historical perspective" (my emphasis).

377    It is true that an effect of flood experience is commonly explained by the availability heuristic, but flood experience was used as an explanatory variable for risk perception and hazard adjustment much earlier than Tversky and Kahneman (1974) proposed the availability heuristic (e.g., Kates, 1963). More generally, Wyer & Albarracín (2005) expanded on the concept of availability by noting that the four factors influencing the retrieval of belief-relevant knowledge are recency, frequency, strength of association, and schematic processing. More recently, Demuth (2018) proposed a comprehensive method of measuring experience.

404    Research hypotheses (e.g., Var1 is positively related to Var2) are one important implication of a theory, but research questions (Is Var1 significantly related to Var2?) can also make a significant contribution to the research literature. Research questions are especially relevant when there are conflicting results in the research literature (e.g., a mixture of significant positive, nonsignificant, and possibly even significant negative relations) that preclude a research hypothesis.

406    Just as important as the derivation of constructs from a theoretical framework is the operationalization of those constructs in terms of indicators (e.g., questionnaire items) that are either consistent with previous operationalizations or are derived from a different theoretical orientation. For example, American disaster researchers define risk perception in terms of personal consequences (e.g., Mileti et al., 1992) that are distinctly different from the risk dimensions in Slovic's (1987) framework. As suggested above, inconsistency in research findings can often be attributed to differences in the operationalization of constructs such as experience and risk perception (Lindell & Perry, 2000).

409    One nonobvious implication of Question 6 is the need for researchers to report the interrelationships among all of the variables that have been measured to test their research hypotheses. Unfortunately, some researchers report only the variables that have significant correlation/regression coefficients with a study's dependent variable(s) such as risk perception and behavior/behavioral intentions. This is a variant of the "file drawer problem" (Rosenthal, 1979). This mistaken practice distorts the scientific record by inflating the estimated effect sizes for the reported variables in statistical meta analyses because only larger, statistically significant, effect sizes are reported in the literature.

425    One limitation to the recommendation for emphasizing the "currently most frequently used

questions and explanatory variables" is that Figure 11 shows that these are demographic variables, which have been found to have small and inconsistent effects, as least in the case of hurricane evacuation (Baker, 1991; Huang et al., 2016). Moreover, this conclusion is consistent with Fishbein and Ajzen's (1975) proposal that demographic variables affect behavior *indirectly* through their effects on psychological variables. Conversely, the attributes of protective actions do not appear in Figure 11's variable list at all despite Fishbein and Ajzen's proposal that an attitude toward an object (i.e., risk perception) is less predictive of behavior than attitudes toward actions related to that object (protective actions/adaptive behaviors). In PMT, the attributes of the protective actions are response-efficacy and response costs (e.g., Floyd et al., 2000). In the PADM (Lindell, 2018; Lindell & Perry, 2012), they are hazard-related attributes (protection of persons and property, utility for other purposes) and resource-related attributes (cost, and requirements for knowledge/skill, time/effort, tools/equipment, and social cooperation).

439     The issue of regional differences can be addressed very effectively in statistical meta analyses.

469     The term "temporal and analysis scale variation" is unclear and needs to be defined.

*References*

Baker, E. J. (1991). Hurricane evacuation behavior. *International Journal of Mass Emergencies and Disasters*, *9*(2), 287-310.

Clarke, L., & Chess, C. (2008). Elites and panic: More to fear than fear itself. *Social Forces*, *87*(2), 993-1014.

Cohen, M. D., March, J. G., & Olsen, J. P. (1972). A garbage can model of organizational choice. *Administrative Science Quarterly*, *17*(1), 1-25.

Demuth, J. L. (2018). Explicating experience: Development of a valid scale of past hazard experience for tornadoes. *Risk Analysis*, *38*(9), 1921-1943.

Fishbein, M., & Ajzen, I. (1975). *Belief, Attitudes, Intention, and Behavior: An Introduction to Theory and Research*. Reading, MA: Addison Wesley Publishing.

Floyd, D. L., Prentice-Dunn, S., & Rogers, R. W. (2000). A meta-analysis of research on protection motivation theory. *Journal of Applied Social Psychology*, *30*(2), 407-429.

Fritz, C. E., & Marks, E. S. (1954). The NORC studies of human behavior in disaster. *Journal of Social Issues*, *10,* 26-41.

Huang, S. K., Lindell, M. K., & Prater, C. S. (2016). Who leaves and who stays? A review and statistical meta-analysis of hurricane evacuation studies. *Environment and Behavior*, *48*(8), 991-1029.

Kates, R. W. (1963, December). Perceptual regions and regional perception in flood plain management. In *Papers of the Regional Science Association* (Vol. 11, No. 1, pp. 215-227). Springer-Verlag.

Lindell, M.K. (2018). Communicating imminent risk. In H. Rodríguez, J. Trainor, and W. Donner (eds.) *Handbook of Disaster Research, 2nd ed.* (pp. 449-477)*.* New York: Springer.

Lindell, M.K. & Perry, R.W. (2000). Household adjustment to earthquake hazard: A review of research. *Environment and Behavior, 32*(4)*,* 461-501.

Lindell, M.K. & Perry, R.W. (2012). The Protective Action Decision Model: Theoretical modifications and additional evidence. *Risk Analysis, 32*(4)*,* 616-632.

Lindell, M.K. & Prater, C.S. (2000). Household adoption of seismic hazard adjustments: A comparison of residents in two states. *International Journal of Mass Emergencies and Disasters, 18*(2)*,* 317-338.

Mileti, D. S., Fitzpatrick, C., & Farhar, B. C. (1992). Fostering public preparations for natural hazards: Lessons from the Parkfield earthquake prediction. *Environment*, *34*(3), 16-39.

Moors, A., & De Houwer, J. (2007). What is automaticity: An analysis of its component features and their interrelations. In J. A. Bargh (ed.) *Social Psychology and the Unconscious* (pp. 11-50)*.* New York: Psychology Press.

Rosenthal, R. (1979). The "file drawer problem" and tolerance for null results. *Psychological Bulletin*, *86*(3), 638-641.

Shrout, P. E., & Rodgers, J. L. (2018). Psychology, science, and knowledge construction: Broadening perspectives from the replication crisis. *Annual Review of Psychology*, *69*, 487-510.

Slovic, P. (1987). Perception of risk. *Science*, *236*(4799), 280-285.

Wyer Jr., R. S., & Albarracín, D. (2005). Belief formation, organization, and change: cognitive and motivational influences. In Albarracín, D., Johnson, B. T., & Zanna, M. P. (eds). *The Handbook of Attitudes* (pp. 273-322). Mahwah NJ: Erlbaum.

---

## Referee Comment (RC2)

**Article review: Rufat et al., - Surveying the surveyors to address risk perception and adaptive behaviour across cross-study comparability.**

**Overall assessment**

This paper seeks to review how risk perception studies are conducted and which theoretical frameworks are applied across a range of disciplines. As the paper states, current risk perception studies adopt a variety of theoretical frameworks (or none at all), meaning they become difficult to compare and contrast. This paper seeks to assess the extent to which this occurs, and to present some recommendations and suggestions to increase replicability and enable cross-validation. The study adopted the use of a survey distributed to the risk perception research community, with 150 responses. The participants were largely from Europe (the target area of the study) and with a strong skew towards those working in flood risk. The study presents a series of trends drawn from the survey responses, including the principle domains explored in perception studies, how explanatory variables are sought and used to explain context, and the effect of regional variance on responses. At the end of the paper, the authors present a series recommendations on how replicability can be enhanced in the field between studies.

The paper is both interesting to read and presents a step forward in increasing the robustness of studies in the field of risk perception. The piece is well written and structured, and presents a clear progressive narrative through the research. I recommend this piece for publication, subject to some minor corrections, detailed below.

**General comments**

- The introduction, for me, doesn't hit on the significance of comparability of studies enough. It's quite a brief introduction and could be expanded to demonstrate the importance of this study more. E.g. why is comparability and transferability important? What is the impact of not doing this? Why is there a crisis of comparability? And, how is that played out in the broader risk perception literature?

- Much of the literature referred to in the text, and indeed the text in Section 8.1 linking the significance of this research to the existing literature focuses on flood risk perception. This suggests that the breadth of the risk perception literature is not as broad as it could be. Also, with 8.1 so heavily linking the study to flood risk perception literature, I'm surprised to not see this mentioned in the introduction at all. It currently feels like it started as a flood risk perception paper and was turned into something more broad, and this needs to be cleaned a bit to be stronger either one way or the other. I'd expect either some acknowledgement of the potential bias towards flood risk perceptions or a heavier inclusion of other risk perception literature to ensure this is truly representative across the field of risk perception.

- Additionally to the comment made above, the majority of respondents (61%) of participants studied flood risk perceptions, over other types of risks. This suggests a bias in your participants perhaps as a result of the snowball sampling method which should be acknowledged in the text. Also a consideration should be made of whether you think your sample is representative of the field.

- Your study looks at the different risk domains that people study but does not explore the severity of the risks they cover. I wonder if you think the severity of the risk studied (e.g. a risk that may cause damage to infrastructure vs. a risk that has a high probability of loss of life), may change the methods employed to assess the perceptions of those risks?

- Section 7 – not sure if you collected the data, but would have been interesting to have gauged the level of significance of the impact of COVID-19 on perception research, rather than just a yes/no response. Also, a lot of the responses are of the impact on researchers directly, but did any respondents detail a change in their approach or methods to their study designs?

**More specific comments**

- 82 - "This in turn, hinders comparability and transferability… and hampers recommendations for and risk management" – how so? Would like to see an extra sentence or two on the actual impact that the lack of comparability might have.

- 124 – Here you describe interdisciplinary researchers as a minority at 28%, but that's higher than the number of people that identified as geographers (your highest group) at 25%.

- Figure 6 – some of these categories seem to overlap? For example, 'from the literature' and 'from other studies' and 'discussions with colleagues from my field' must have significant overlaps with 'discussions with people in my country' and 'discussions with people in different countries'. Can you make the differentiation of categories clearer?

- 248 – "Socio-demographic characteristics…." Needs reference(s).

- Figure 8 – Might be a formatting thing in review, but the title of the left graph has been cut slightly.

- 458 – Can you add the figure in here for "As a substantial share of studies…".

---

## Author Response (AR1)

**NHESS-2021-365 v3**

Dear editor,

Hereby we resubmit our revised paper entitled *"Surveying the Surveyors to Address Risk Perception and Adaptive Behaviour Cross-study Comparability"* for publication in NHESS. The manuscript is an original piece of work that has not been submitted or published elsewhere.

We appreciate the positive evaluation of our paper and we have answered in the discussion to all questions raised by the reviewers. We have thoroughly reviewed our manuscript to implement the suggestions from the two reviewers as detailed in our answers.

We have revised the Introduction (Section 1) to more clearly explain the objectives of the paper and the significance of comparability of studies. We have expanded the Methods (section 2) and Discussion (Section 8) to further discuss the potential bias towards flood risk perceptions. In Section 4, we have further explained the differences between categories when they seem to overlap while clarifying that they are derived from multiple choice questions. In Section 5, we have expanded the discussion on the reasons for the prevalence of the demographic variables, revised Figure 9 by conducting a hierarchical cluster analysis and made more explicit the suggestion to survey communities that differ substantially in their hazard experience to obtain the requisite variation in experience at the household level. In Section 6, we have revised the statement on sample size and 'high data scattering'; we have also clarified that 'this observation is blurred' remains a speculative explanation. In section 8, we have included a broader risk perception literature to ensure this is more representative across the whole field of risk perception. We made other minor changes at the request of the reviewers, they are detailed in our answer in the discussion. When implementing in the manuscript substantial contributions from the reviewers, we have chosen to refer to their suggestions published in the discussion in NHESS-D.

We hope that our revised version is now ready for publication in NHESS.

Yours sincerely,
Samuel Rufat, on the behalf of all the authors

Answer to R1: Michael Lindell

We appreciate the positive evaluation our paper and would like to thank you for your helpful and comprehensive suggestions for improving it. We are particularly grateful to you for sharing that based on your own extensive experience the data are likely to generalize to other hazards and countries. Thank you also for suggesting additional references to help include a broader risk perception literature, we have implemented them throughout the manuscript and in particular in the Discussion (Section 8) to increase the representativeness across hazards. We have expanded the Introduction (Section 1) to distinguish between inconsistent findings and contradictory findings while more clearly explaining the significance of comparability of studies in particular with respect

to replication and meta-analyses. We have also clarified that the 'minimal requirements' are detailed and discussed in Section 8. In Figure 4, we report the categories as they were collected during the survey. While we comment in Section 4 that grouping some items might change the ranking, we also demonstrate – for example – that respondents do not equate worry with fear. As a result, we feel more comfortable to report in Figure 4 the results as they were collected during the survey and leave interpretation as open as possible to foster discussion. We took however the liberty of adding your analysis of Figure 4 in Section 4, thank you for suggesting this categorization based on theoretical distinctions. Similarly, in Figure 6 and Figure 7, we report the categories as they were collected during the survey. We have clarified that they are based on multiple choice questions. Although some categories may appear to overlap, we have clarified the difference between 'from the literature' and 'from other studies'. Which also led us to clarify that while the majority of respondents reported relying on literature reviews (in general) only a minority considered direct comparison with previous studies (tables and data). In retrospect we realize that our attempt at humour in the title of Section 5 was misplaced and we have refrained from using the term 'panic' to avoid confusion. Thank you for pointing this out. In Section 5, we have expanded on the link you suggested between research experience and coordination efforts. In our sample, we had 44% of respondents with less than 3 studies and 21% with more than 5 studies (Table 1). We further took the liberty of adding your suggestion of another possible reasons for the prevalence of the demographic variables, thank you for highlighting that their inclusion requires no knowledge of the theoretical perspectives we collected. We have also added your recommendation to survey communities that differ substantially in their hazard experience to obtain the requisite variation in experience at the household level – this was implicitly one of our main goals in fostering cross-study comparability. Based on your suggestion, we have revised Figure 9 by conducting a hierarchical cluster analysis and then rearranging the variables in the resulting clusters. As for Figure 11, we have clarified that it is based on the questions presented in Figure 8. Similarly, we clarified the comparison of contextual factors use between interviews and survey designs. In Section 6, we have revised the statement on sample size and 'high data scattering'. We have clarified that we did find regional difference in risk perception and behavior assessment between different regions of Europe. We have also clarified that 'this observation is blurred' remains a speculative explanation. In section 8, we have clarified that flood experience was used as an explanatory variable for risk perception and hazard adjustment earlier than the introduction of the availability heuristic. We have also implemented your other suggestions and references, as well as your sensible point on research questions, thank you for bringing all these up. We have further clarified that beyond the derivation of constructs from a theoretical framework we are also raising the issue of the operationalization of those constructs in terms of indicators. Thank you for bringing up the 'file drawer problem', we took the liberty of adding this to our discussion (Section 8.2). Thank you for highlighting one limitation to the recommendation for emphasizing the currently most frequently used questions and explanatory variables. We have clarified that – as the results also reveal that one third of surveyors did not rely on a particular theoretical model or framework to guide their design – more systematic efforts must be made to integrate the constructs from the main frameworks beyond the currently most frequently used questions and variables. As for your final point that the issue of regional differences can be addressed very effectively in statistical meta-analyses, we have clarified that this is indeed one of the aims of the discussion we hope this study has contributed to launch: we need more convergence between studies to improve comparability to

enable robust and comprehensive statistical meta-analyses. Thank you again for your comprehensive and insightful suggestions which helped considerably improving the paper.

Answer to R2: Lara Mani

We appreciate the positive evaluation our paper and we thank you for your helpful suggestions for improving it. We have revised the Introduction (Section 1) to more clearly explain the objectives of the paper and the significance of comparability of studies in particular with respect to replication and meta-analyses. We have expanded the Methods (section 2) and Discussion (Section 8) to further discuss the potential bias towards flood risk perceptions. We take confidence however in R1's (Michael Lindell) comment on this particular point: "In the absence of a census of researchers and hazards in the field of risk perception and adaptive behavior, it is unknown whether these sample characteristics are representative of the field's researchers. Nonetheless, the authors have identified and summarized problems that I have seen repeatedly during my 50 years studying floods, volcanic eruptions, hazardous materials releases, earthquakes, hurricanes, tornadoes, and tsunamis in the US and other Pacific Basin countries. Consequently, their data are likely to generalize to other hazards and countries." We have also followed Michael Lindell's comprehensive suggestions to help include a broader risk perception literature throughout the manuscript and in particular in the Discussion (Section 8) to ensure this is more representative across the whole field of risk perception. For Figure 6 (Section 4), we have clarified that it is based on a multiple choice question. Even though some categories may appear to overlap, we have clarified the difference between 'from the literature' and 'from other studies', as for 'colleagues in my field' and 'people in different countries' they were designed to capture different behaviour. As for the impact of COVID-19 on perception research, this was an open and optional question in the survey. As reported in Section 7, some colleagues did detail a change in their approach, methods or study designs – mostly online interviews and online surveys – fewer, however, than those who reported having to postpone their studies. We do not feel their small numbers allow for a more detailed estimation or quantification of the impact of the pandemic on perception research. Based on your other comments, we have made other ad hoc clarifications in Section 2 (interdisciplinary), Section 5 (socio-demographic characteristics), Figure 8 (formatting) and Section 8 (share of studies). Thank you again for your detailed and useful comments which helped improving the paper.